# INTERPRETABLE DEEP CLUSTERING

## ABSTRACT

Clustering is a fundamental learning task widely used as a first step in data analysis. For example, biologists use cluster assignments to analyze genome sequences, medical records, or images. Since downstream analysis is typically performed at the cluster level, practitioners seek reliable and interpretable clustering models. We propose a new deep-learning framework for tabular data that predicts interpretable cluster assignments at the instance and cluster levels. First, we present a self-supervised procedure to identify the subset of the most informative features from each data point. Then, we design a model that predicts cluster assignments and a gate matrix that provides cluster-level feature selection. Overall, our model provides cluster assignments with an indication of the driving feature for each sample and each cluster. We show that the proposed method can reliably predict cluster assignments in synthetic and tabular biological datasets. Furthermore, using previously proposed metrics, we verify that our model leads to interpretable results at a sample and cluster level.

## 1 INTRODUCTION

Clustering is an essential task in data science that enables researchers to discover and analyze latent structures in complex data. By grouping related data points into clusters, researchers can gain insights into the underlying characteristics of the data and identify relationships between samples and variables. Clustering is used in various scientific fields, including biology (Reddy et al., 2018), physics (Mikuni & Canelli, 2021), and social sciences (Varghese et al., 2010). For example, in biology, clustering can identify different disease subtypes based on molecular or genetic data. In psychology, based on survey data, clustering can identify different types of behavior or personality traits.

One of the most common applications of clustering in bio-medicine is the analysis of gene expression data, where clustering can be used to identify groups of genes with similar expression patterns across different samples (Armingol et al., 2021). Scientists are often interested in clustering the high-dimensional points corresponding to individual cells, ideally recovering known cell populations while discovering new and perhaps rare cell types (Deprez et al., 2020). Bio-med gene expression data is generally represented in tabular high-dimensional format, thus making it difficult to obtain accurate clusters with meaningful structures. In addition, interpretability is a crucial requirement for real-world bio-med datasets where it is vital to understand the biological meaning behind the identified clusters (Yang et al., 2021). Therefore, there is an increasing demand in biomedicine for clustering models that offer interpretability for tabular data.

In recent years, there has been a growing interest in deep learning models for clustering (Shen et al., 2021; Li et al., 2022; Cai et al., 2022; Niu et al., 2021). The neural network offers an improved embedding of data points, thus raising the bar of clustering capabilities. However, most existing schemes focus on image data, require domain-specific augmentations, and are not interpretable. Interpretability has also been gaining attention in deep learning, but most models focus on supervised learning Alvarez Melis & Jaakkola (2018); Yoon et al. (2019); Yang et al. (2022). We aim to generalize these ideas to unsupervised learning by designing a deep clustering model for tabular (biomedical) data that is interpretable by design. Here, we consider interpretability as the ability to identify variables that *drive* the formation of clusters in the data (Bertsimas et al., 2021).

This work presents Interpretable Deep Clustering (IDC), an unsupervised two-stage clustering method for tabular data that first selects samples-specific features that are informative for reliable data

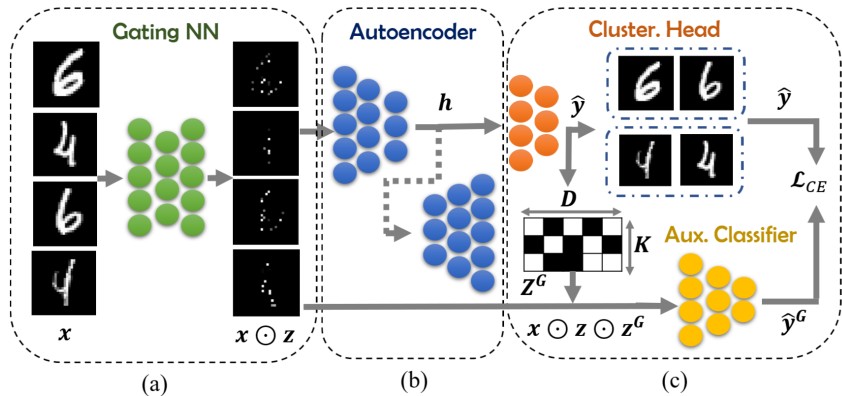

Figure 1: Illustration of the proposed model. The first step involves self-supervision for learning a meaningful latent representation and sample-level informative features. During this stage, we optimize the parameters of the Gating Network (green) and the autoencoder (blue) that reconstructs $\hat{x}$ from latent embedding $h$. The gating Network learns a sample-specific sparse gate vector $z$ for input sample $x$ such that $x \odot z$ is sufficient for reconstruction via an autoencoder. During the second stage, we train a clustering head (orange) to predict cluster assignments $\hat{y}$ from the latent embedding $h$ by minimizing the mean cluster coding rate loss (see Eq. 6). This loss is designed to push clusters apart while making each cluster more compact. The Auxiliary Classifier (yellow) is trained on sparse representations $x \odot z \odot Z^G$ to predict cluster labels and optimizes the cluster level gating matrix $Z^G$.

reconstruction– a task that is known to be correlated with clustering capabilities (Song et al., 2013; Han et al., 2018). Then, using the sparsified data, we learn the cluster assignments by optimizing neural network parameters subject to a clustering objective function Yu et al. (2020). In addition, the method provides both instance-level and cluster-level explanations represented by the selected feature set. The model learns instance-level *local gates* that select a subset of features using an autoencoder (AE) trained to reconstruct the original sample. The *global gates* for cluster-level interpretability are derived from the cluster label assignments learned by the model. To enforce sample-level interpretations, the gates are encouraged by the recently proposed discrimination constraint denoted as the total coding rate. Using synthetic data and MNIST we demonstrate the interpretability quality of our model. Then, using real-world tabular data we demonstrate that our model can find reliable clusters while using only on a small subset of informative features. In the following sections, we provide a detailed description of our approach.

## 2 RELATED WORK

**Unsupervised Feature Selection** The problem of unsupervised feature selection (UFS) involves identifying variables useful for downstream tasks such as data clustering. Towards this goal, several works have exploited regularized AE (Han et al., 2018; Lee et al., 2022; Sokar et al., 2022; Balın et al., 2019) that identify a global subset of features sufficient for data reconstruction. Another line of UFS schemes relies on the graph Laplacian to identify subsets of smooth features with respect to the core structure in the data (He et al., 2005; Zhao & Liu, 2012; Lindenbaum et al., 2021; Shaham et al., 2022). Both types of UFS frameworks can help improve downstream clustering capabilities; however, existing global schemes do not provide sample-level or cluster-level interpretability. While there are recent works on supervised local feature selection (Yoon et al., 2019; Yang et al., 2022) that provide interpretability, we are not aware of any sample-level unsupervised feature selection schemes. Here, we present an end-to-end clustering scheme with local feature selection capabilities for the first time.

**Interpretable clustering** Guan et al. (2011) presented a pioneering work for applying unsupervised feature selection and clustering. The authors proposed a probabilistic model that performs feature selection by using beta-Bernoulli prior in the context of a Dirichlet process mixture for clustering. The method can only select cluster-level and dataset-level informative features; in contrast, our method provides interpretability for sample-level granularity. In (Frost et al., 2020), the authors proposed tree-based $K$-means clustering. However, the explanations are static for a given dataset and rely on the whole set of points. In contrast, our approach learns local gates for each sample in the dataset by optimizing the neural network as a gate selector, thus producing sample-level interpretations. Since our method is fully parametric, it also offers generalization capabilities compared to existing schemes.

Specifically, our model can predict cluster assignments and informative features for samples not seen during training.

**Deep clustering**    Recently, several methods were proposed for NN-based clustering, to name a few: (Gao et al., 2020; Niu et al., 2021; Li et al., 2022; Shaham et al., 2018; Cai et al., 2022; Lv et al., 2021; Shen et al., 2021; Peng et al., 2022). However, the primary domain of these works is vision, and they rely on domain-specific augmentations and, therefore, can not be applied to tabular datasets. Here, we introduce an interpretable NN model for general high-dimensional datasets such as bio-med tabular data. Li et al. (2022) have generalized the maximum coding rate reduction loss (MCR$^2$) (Yu et al., 2020) for embedding and clustering. The model seeks to push clusters apart while making them denser. One caveat of the approach is that it requires dataset-specific augmentation for learning a latent embedding using a contrastive loss. While such augmentations could be efficiently designed for the visual domain, semantic preserving augmentations of tabular data remain an open challenge (Qian et al., 2023). To overcome this limitation, we propose to learn the sparse representation of a sample to retain essential feature variables for clustering via a two-step procedure. The first involves self-supervision with locally sparse reconstruction, and the second adapts the MCR$^2$ (Li et al., 2022) for identifying clusters based on diverse features.

## 3    PROBLEM SETUP

We are interested in clustering data points $\mathbf{X} = \{\boldsymbol{x}_i\}_{i=1}^N$ into matching clusters $\mathbf{Y} = \{\boldsymbol{y}_i\}_{i=1}^N$, where $\mathbf{x}_i \in R^D$ are $D$-dimensional vector-valued observations of general type, i.e., tabular that do not obey a particular feature structure. Our goal is to learn an interpretable clustering model defined by the tuple $\langle f_\Theta, \mathcal{S}^{glob} \rangle$ such that $f_\Theta(\boldsymbol{x}_i) = \{\hat{y}_i, \mathcal{S}_i^{loc}\}$ where $\hat{y}_i \in \{1, 2, ..., K\}$ is an accurate clustering assignment, and $\mathcal{S}_i^{loc}$ is a local feature importance set for sample $i$ and defined by $\mathcal{S}_i^{loc} = \{s_i^j \in [0, 1]\}_{j=1}^D$. $\mathcal{S}^{glob} \in [0, 1]^{K \times D}$ is a global feature importance matrix where each gating vector of size $D$ is learned for $K$ clusters. By forcing $|\mathcal{S}_i^{loc}| << D$, we can attenuate nuisance features and identify (sample-specific) subsets of informative features, thus improving the interpretability of the clustering model.

Our work is motivated by the vital task in biomedicine of cell clustering with marker gene identification (Kiselev et al., 2017). The task involves unsupervised clustering of a high-dimensional dataset while identifying genes with unique patterns in each cluster. Therefore, we aim to design an interpretable model that performs well in the clustering task, i.e., can identify groups with semantically related samples while focusing on local subsets of features. We note that in the supervised setting, this type of local feature selection was tied to interpretability by several authors (Alvarez Melis & Jaakkola, 2018; Yoon et al., 2019; Yang et al., 2022).

We generalize these ideas of supervised interpretability to the unsupervised setting and propose a novel model that performs clustering and local feature selection. The model learns to exclude nuisance input features that do not contribute to the clustering learning task. In addition, local feature selection produces unique explanations for each sample in addition to cluster-level interpretations, which are meaningful for understanding the model predictions. We use several metrics to demonstrate that our model improves interpretability while leading to accurate clustering results.

Our model is trained without access to any labeled data. In the evaluation step, we only use labels to evaluate the quality of our model in terms of feature selection and clustering by measuring the clustering Accuracy (ACC) based on ground truth categories $y_i$ (labels).

## 4    METHOD

### 4.1    INTUITION

We propose a two-step framework that first learns sample-level gates and latent representation by self-supervised training. The model learns both *local gates* for interpretable input space representation and latent embedding to better discriminate between samples. In the second step, we learn the cluster assignments and global gates, highlighting cluster-level driving features. The *global gates* are learned based on the clustering assignments, which are recovered by maximizing the sum of *coding rate*

for each cluster. Finally, the method produces clustering assignments for each sample, local gates for sample-level interpretation, and global gates for the cluster-level explanation. In the following subsection, we present the proposed framework and architecture design.

## 4.2 LOCAL SELF SUPERVISED FEATURE SELECTION

Given unlabeled observations $\{\mathbf{x}_i\}_{i=1}^N$ we want to learn a prediction function $f_\theta$ (parametrized using a NN) and sets of indicator vectors $\mathbf{s}_i \in \{0,1\}^D$ that will "highlight" which subset of variables the model should rely on for clustering each sample $\mathbf{x}_i$. This will enable the model to attain fewer features for each sample and reduce overfitting when predicting the clusters of unseen samples.

Towards this goal, we extend the recently proposed stochastic gates (Lindenbaum et al., 2021; Yamada et al., 2020) to learn the indicator vector. Stochastic gates are continuously relaxed Bernoulli variablesdefined here (for feature $d$ and sample $i$) based on the following hard thresholding function

$$\boldsymbol{z}_i = \max(0, \min(1, 0.5 + \boldsymbol{\mu}_i + \boldsymbol{\epsilon}_i)), \tag{1}$$

where $\boldsymbol{\epsilon}_i$ is drawn from $\mathcal{N}(0, \sigma^2)$, and $\boldsymbol{\mu}_i$ are the network output logits before the hard thresholding function (in Eq. 1), $\sigma$ is fixed throughout training to $0.5$ in our model as suggested in (Yamada et al., 2020) and controls the injected noise. The role of the injected noise is to push the converged values of $\boldsymbol{z}_i$ towards 0 or 1; see (Yamada et al., 2020; Yang et al., 2022) for further details. The sample-specific parameters $\boldsymbol{z}_i \in \mathbb{R}^D, i = 1, ..., N$ are predicted based on a *gating* network $f_{\theta_1}$ such that $\boldsymbol{\mu}_i = f(\boldsymbol{x}_i | \theta_1)$, where $\theta_1$ are the weights of the *gating* network. These weights are learned simultaneously with the weights of the *prediction* network $\theta_2$ by minimizing the following loss:

$$\mathcal{L}_{sparse} = \mathbb{E}\big[\mathcal{L}(f_{\theta_2}(\boldsymbol{x}_i \odot \boldsymbol{z}_i)) + \lambda_t \cdot \mathcal{L}_{reg}(\boldsymbol{z}_i)\big], \tag{2}$$

where $\mathcal{L}$ is a desired prediction loss, e.g., clustering objective function or reconstruction error. The Hadamard product (element-wise multiplication) is denoted by, and we compute the empirical expectation over $\boldsymbol{x}_i$ and $\boldsymbol{z}_i$, for $i$ in a dataset of size $N$. The term $\mathcal{L}_{reg}(\boldsymbol{z}_i)$ is a regularizer that is designed to sparsify the gates and is defined by: $\mathcal{L}_{reg} = \|\boldsymbol{z}_i\|_0$. After taking the expectation (over $\boldsymbol{z}_i$ and the samples $\boldsymbol{x}_i$), $\mathbb{E}[\mathcal{L}_{reg}]$ can be rewritten using a double sum in terms of the Gaussian error function (erf):

$$\mathbb{E}[\mathcal{L}_{reg}] = \frac{1}{N} \sum_{i=1}^N \sum_{d=1}^D \left(\frac{1}{2} - \frac{1}{2} \operatorname{erf}\left(-\frac{\mu_i^d + 0.5}{\sqrt{2}\sigma}\right)\right), \tag{3}$$

here, we take the expectation using the parametric definition of $\boldsymbol{z}_i$.

We pick a denoising autoencoder (Vincent et al., 2008) as our prediction network to select only informative features for clustering and disregard nuisance features. By training with self-supervision using a reconstruction loss with augmentation, the network learns a latent embedding of the input sample and drives the gating network to open gates only for features required to reconstruct data. The model consists of the following:

- *Gating Network*: $f_{\theta_G}(\boldsymbol{x}_i) = \boldsymbol{z}_i$, is a hypernetwork that predicts the gates $\boldsymbol{z}_i$ vector for sample $\boldsymbol{x}_i$, where $\boldsymbol{z}_i \in [0,1]^D$.
- *Encoder*: $f_{\theta_E}(\boldsymbol{x}_i') = \boldsymbol{h}_i$, is a mapping function that learns an embedding $\boldsymbol{h}_i$ based on the element-wise gated sample $\boldsymbol{x}_i' = \boldsymbol{x}_i \odot \boldsymbol{z}_i$.
- *Decoder*: $f_{\theta_D}(\boldsymbol{h}_i) = \hat{\boldsymbol{x}}_i$, that reconstructs $\boldsymbol{x}_i$ based on the embedding $\boldsymbol{h}_i$.

We train autoencoder parametrized by $\theta_2 = \theta_E \cup \theta_D$ with gated input reconstruction loss $\mathcal{L}_{recon}(f_{\theta_2}(f_{\theta_1}(\boldsymbol{x}_i) \odot \boldsymbol{x}_i), \boldsymbol{x}_i)$, which measures the deviation of estimated $\hat{\boldsymbol{x}}_i = f_{\theta_2}(f_{\theta_1}(\boldsymbol{x}_i) \odot \boldsymbol{x}_i)$ from the input sample $\boldsymbol{x}_i$. We introduce input (Vincent et al., 2008) and latent (Doi et al., 2007) data augmentations to learn semantically informative features. Additional details about the augmentations appear in Appendix S7. In addition, we introduce an additional *gates total coding loss*, $\mathcal{L}_{gtcr}(\boldsymbol{z}_i)$, that encourages the model to select unique gates for each sample and is defined by the equation:

$$\mathcal{L}_{gtcr} = -\mathbb{E}_{\mathbb{Z}}\big[\frac{1}{2} \cdot \operatorname{logdet}(\mathbf{I} + \lambda_1 \cdot (\boldsymbol{z}_i^T \boldsymbol{z}_i))\big], \tag{4}$$

which is approximately the negative Shannon coding rate of a multivariate Gaussian distribution (Yu et al., 2020), and is defined for a vector of local gates $\mathbf{z}_i$. This component, inspired by (Li et al.,

2022), is designed to push the model to increase the amount of information represented in a $z_i$. $\lambda_1$ is a constant; we describe how it is tuned in the appendix.

The final sparse prediction loss is $\mathcal{L}_{sparse} = \mathcal{L}_{recon} + \mathcal{L}_{gtcr} + \lambda(t) \cdot \mathcal{L}_{reg}$, where the regularization weight term $\lambda(t)$ is increased during the training using a cosine function scheduler. The weights of the autoencoder are initialized by first training with an $|| \cdot ||_1$ reconstruction loss between $x_i$ and $\hat{x}_i = f_{\theta_2}(x_i)$. In the ablation study presented in Section 6.2.5, we corroborate the importance of each component of our loss.

Equipped with this loss, our model sparsifies the input samples to the minimum number of features that contain the required information for the data reconstruction. While input denoising reconstruction loss applies a random mask during the learning, the gated input reconstruction loss enables learning local masks, or gates, that attenuate noisy features and improve the interpretability of the model. The gates' total coding loss encourages the selected gates to be more diverse, thus providing unique gates for each sample.

### 4.3 CLUSTER ASSIGNMENTS WITH GLOBAL INTERPRETATIONS

**Clustering Head** In the clustering phase, we aim to compress the learned representations of gated samples of $\mathbf{X}$ into $K$ clusters and predict cluster-level gates. To achieve that, we train a clustering head $f_{\theta_3}(h_i) = \hat{y}_i$, which learns cluster one-hot assignments $\hat{y}_i \subset \{1, ..., K\}$. The model outputs logits values $\{\pi_i^k\}_{k=1}^K$ for each sample which are converted to cluster assignment probabilities $\hat{y}_i$ by the Gumbel-Softmax (Jang et al., 2016) reparameterization:

$$\hat{y}_i^k = \frac{\exp((\log(\pi_i^k) + g_i^k)/\tau)}{\sum_{j=1}^K \exp((\log(\pi_i^k) + g_i^k)/\tau)}, \qquad \text{for } k = 1, ..., K. \tag{5}$$

where $\{g_i^k\}_{k=1}^K$ are i.i.d. samples drawn from a Gumbel(0, 1) distribution, and $\tau$ is a temperature hyper parameter. High values of $\tau$ produce a uniform distribution of $\hat{y}$ while decreasing the temperature yields one-hot vectors. This neural network is optimized with the following loss

$$\mathcal{L}_{head} = \sum_{k=1}^K \frac{1}{2} \cdot \text{logdet}\big[\mathbf{I} + \lambda_2 \cdot \mathbb{E}_{h_i \in \mathbf{H}_k}(h_i^T h_i)\big], \tag{6}$$

where $\mathbf{H}_k = \{f_{\theta_E}(x_i \odot z_i) = h_i\}_{i=1}^{B^k}$ are embedding vectors for all samples $x_i$'s which were assigned to cluster $k = \arg\max_K(\hat{y}_i)$, and $B^k$ is the size of the cluster. In contrast to 7, here we would like to decrease the coding rate on average for each cluster of embeddings $\mathbf{H}_k$ to make the clusters more compact. The embeddings are not optimized during this step but only cluster assignments.

**Global Interpretations** We propose an extension to the model that provides cluster-level interpretations by training a *Global Gates Matrix*, $\mathbf{Z}_G \in \{0, 1\}^{K \times D}$ where each row $z_G^k$ corresponds to a cluster and each column to an input variable. To train this gate matrix, we utilize an *Auxiliary Classifier*, $f_{\theta_4}$, that accepts a gated representation of $x_i$ which is defined by $x_i \odot z_i \odot z_G^k$, and $z_G^k$ is a global gates vector learned for cluster $k = \text{argmax}_{1,...,K} \hat{y}_i$. The locally sparse samples $x_i \odot z_i$ learned with autoencoder are multiplied by global gates $\mathbf{Z}_G$ and fed into the single-hidden-layer classifier. The classifier is trained to output cluster assignments $\hat{y}_i$ identical to those predicted with clustering head $f_{\theta_4}$ and is optimized with a cross-entropy loss. During the inference, only $\mathbf{Z}_G$ is used and $f_{\theta_4}$ is discarded.

Finally, similarly to the local gates optimization, we use regularization loss term $\mathcal{L}_{reg}$ with increasing weight $\lambda_t^g$, which sparsifies the gates in the global gates matrix: $\mathcal{L}_{clust} = \mathcal{L}_{head} + \mathcal{L}_{CE} + \lambda_t^g \cdot \mathcal{L}_{reg}$. To summarize, we train the clustering head to predict assignments jointly with global gates. An auxiliary classifier optimizes the global gates while being trained in a self-supervised fashion on the pseudo labels predicted by the clustering head.

## 5 INTERPRETABILITY

Practitioners may require interpretability at different levels of granularity. At the coarser level, it is interesting to identify which features are common to a group of semantically related samples

(or clusters) (Guan et al., 2011). At the finer level, we seek unique explanations for each data point, namely what features drive the model to make specific predictions (Alvarez Melis & Jaakkola, 2018;?). While the model's internal functionality may remain a black box to the practitioner, the correlation between the input space of feature values and model predictions sheds some light on the model's interpretability. Several criteria were recently proposed to evaluate the interpretability of supervised models (Alvarez Melis & Jaakkola, 2018; Yang et al., 2022). We present them in the following paragraphs and discuss their modifications to our unsupervised setting.

**Diversity** We expect a good interpretability model to identify different sets of variables as driving factors for explaining distinct clusters. The *diversity* metric measures this quantity and is computed by negative mean Jaccard similarity between cluster-level informative features across all pairs of clusters. Given a set of indices $S_{c_i} \subset 1, ..., D$ indicating the selected informative variables of $c_i, i = 1, ..., K$, the *diversity* is defined as $1 - \sum_{i \neq j} \frac{J(S_{c_i}, S_{c_j})}{K \cdot (K-1)/2}$. Here, $J$ is the Jaccard similarity between two sets, and perfect diversity is obtained at 1, indicating no overlap between cluster-level features.

**Faithfulness** An interpretation is faithful if it accurately represents the reasoning behind the model's prediction function. To evaluate this quantity, *faithfulness* measures the correlation between the predictivity of the model and the feature importance. Specifically, we first compute a feature's importance value, for instance, the value of our corresponding predicted gate averaged over all samples. Then, we sort the vector of importance values, remove features individually (from most to least important), and measure the performance of the clustering model. If the model's performance drops monotonically with feature importance, we will obtain a high correlation, indicating that the prediction model is faithful to the learned feature importance values. An example of this metric is presented in Fig. 3.

**Uniqueness** The motivation for uniqueness is opposite to the *stability* metric proposed in (Yang et al., 2022; Alvarez Melis & Jaakkola, 2018), which requires the selected features to be consistent between close samples. Since we are interested in sample-level interpretation, we extend the *diversity* to a metric that compares samples instead of clusters. Specifically, we propose to measure the *uniqueness* of the selected features for similar samples, or in other words, how granular are our explanations. We define *uniqueness* of the explanations by: $\min_{x_i, x_k \leq \epsilon} \frac{\|w_i - w_k\|_2}{\|x_i - x_k\|_2}$, where $w_i, w_k$ are feature sets for samples $x_i, x_k$. The smaller this value is, the less sample-specific the interpretation of the model is. Therefore, we want our model to obtain high *uniqueness* values.

## 6 EXPERIMENTS

We conducted six types of experiments. First, we verify our method's clustering and local feature selection capabilities on a synthetic dataset. Then, we verify that the sparsity constraint preserves or even improves clustering quality for two commonly used clustering benchmarks, MNIST and FashionMNIST. To test the model's generalization capability, we test the trained model on an unseen test set from the MNIST dataset. In the fourth experiment, we evaluate our method on small sample size high dimensional real-world tabular datasets. Next, we evaluate the interpretability of the model on an MNIST subset of size $10,000$ samples and present the quality of the selected features. Finally, we test the proposed method on a more challenging CIFAR10 dataset with a convolutional neural network (CNN) backbone without additional augmentations. The datasets used in the experiments are summarized in Table S6 in the Appendix. We note that the real-world datasets are still considered challenging by several studies on clustering with tabular data (Shaham et al., 2022; Xu et al., 2023). In addition, we compare the interpretability metrics measured for our method against other popular feature explanation schemes.

### 6.1 EVALUATION

**Interpretability quality** To evaluate the interpretability of the proposed model, we train it on the $\text{MNIST}_{10K}$ images, $1K$ images for each class. We measure diversity, uniqueness, and faithfulness described in Section 5. The evaluation is done on subsets of $1K$ images by taking the mean value after ten iterations. We compare the interpretability of our method to the popular SHAP feature importance detection method (Lundberg & Lee, 2017) implemented here [1] and trained on $K$-Means outputs. In

---

[1] https://github.com/slundberg/shap

Table 1: Evaluating the interpretability quality of our model on the MNIST$_{10K}$ data. Our IDC model improved clustering accuracy. We use it to compare (i) the top features explained by SHAP trained based on a $K$-means model, (ii) Integrated Gradients, and (iii) Gradient SHAP applied as explainers to our model. Our local gating network selects faithful (0.96) and unique (0.69) features while providing comparable diversity values (94.8).

| Method | ACC ↑ | $|\mathcal{S}|$ ↓ | Uniqueness ↑ | Diversity ↑ | Faithfulness ↑ |
|---|---|---|---|---|---|
| $K$-means + SHAP | 53.34 | 15 | 0.12 | **100.0** | 0.79 |
| TELL (Peng et al., 2022) + IntegGrads | 74.79 | 15 | 0.03 | 89.1 | 0.67 |
| TELL (Peng et al., 2022) + GradSHAP | 74.79 | 15 | 0.15 | 92.5 | 0.63 |
| IDC w/o gates + IntegGrads | 82.32 | 15 | 0.02 | 95.8 | 0.78 |
| IDC w/o gates + GradSHAP | 82.32 | 15 | 0.08 | **100.0** | 0.86 |
| IDC + IntegGrads | **83.45** | 15 | 0.01 | 95.3 | 0.94 |
| IDC + GradSHAP | **83.45** | 15 | 0.02 | 97.0 | 0.93 |
| IDC | **83.45** | 15 | **0.69** | 94.8 | **0.96** |

addition we compare interpretability of our method against Gradient SHAP [2] and Integrated Gradients (Sundararajan et al., 2017). Both are trained on IDC clustering model predictions inside Label-free XAI framework (Crabbé & van der Schaar, 2022).

**Clustering accuracy** We use three popular clustering evaluation metrics: Clustering Accuracy (ACC), The adjusted Rand index score (ARI), and Normalized Mutual Information (NMI).

## 6.2 RESULTS

### 6.2.1 SYNTHETIC DATASET

Table 2: Results on the Synthetic dataset with three informative features and ten nuisance features. Our model is accurate and achieves the highest F1-score regarding its ability to select the correct informative features.

| Method | ACC ↑ | F1-score ↑ |
|---|---|---|
| IDC | **99.91** | **88.95** |
| $K$-means+SHAP | 25.72 | 49.65 |

We verify our method first on the synthetic dataset. Inspired by (Armanfard et al., 2015), the dataset consists of three informative features $x_i[j] \in [-1, 1], j = 1, .., 3$ for each sample $\boldsymbol{x}_i$, in which we generate three Gaussian blobs [3]. The detailed description of the dataset generation could be found in I. We add ten nuisance background features, resulting in 13 total features. The samples are equally distributed between 4 clusters, with $\sim 800$ samples in each cluster. Given the first two dimensions $\{x[1], x[2]\}$, only 3 clusters are separable, and the same property holds for dimensions pair $\{x[1], x[3]\}$. We expect the interpretable clustering model to be able to select the correct support features for each cluster. In Table 2, we present the accuracy of the clustering where $K$-means was run 20 times, and the F1-score was measured on the selected features. We expect clusters with purple and blue labels to be explained by features $\{x[1], x[2]\}$ and clusters with green and yellow labels by $\{x[1], x[3]\}$. To evaluate the model's effectiveness, we calculate both clustering accuracy (ACC) and F1-score that measures the quality feature selection. Since for each sample, we know what the informative features are, we can calculate precision and recall for gate-level feature selection. While $K$-means fails to produce accurate clustering, our method achieves $99.91\%$ clustering accuracy. Additionally, it has an excellent ability to select the relevant features. The results are presented in Table 2.

### 6.2.2 INTERPRETABILITY RESULTS ON MNIST$_{10K}$ DATASET

As a baseline for interpretability, we exploit the SHAP values model, which is trained on the $K$-means model predictions. First we train $K$-means on MNIST$_{10K}$ dataset. Since SHAP requires a set of samples used as a background during SHAP training, we select an additional 100 random MNIST images that are not in the train set. Once the SHAP model is trained, we take 15 top feature importance indices, the number of selected gates by our model. As indicated in Table 1, our method produces more expressive features as sample-level explanations with the highest uniqueness score. In addition, our model outperforms SHAP in faithfulness, as presented in Fig. 3. It is easy to see that the accuracy drop correlates with feature importance. Additionally, Fig. 3 presents the active gates selected for each sample. It can be seen that **IDC** selects more informative features that are local for each sample.

---

[2]https://captum.ai/api/gradient_shap.html

[3]https://scikit-learn.org/stable/modules/generated/sklearn.datasets.make_blobs.html

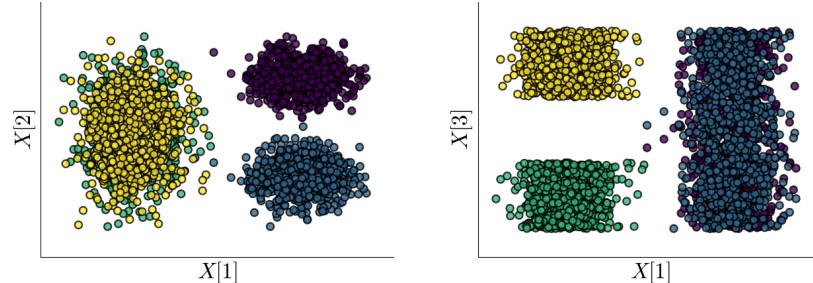

Figure 2: Visualization of Synthetic dataset. To separate between clusters, the model should select one of the pairs $\{x[1], x[2]\}$ or $\{x[1], x[3]\}$ of non-background features.

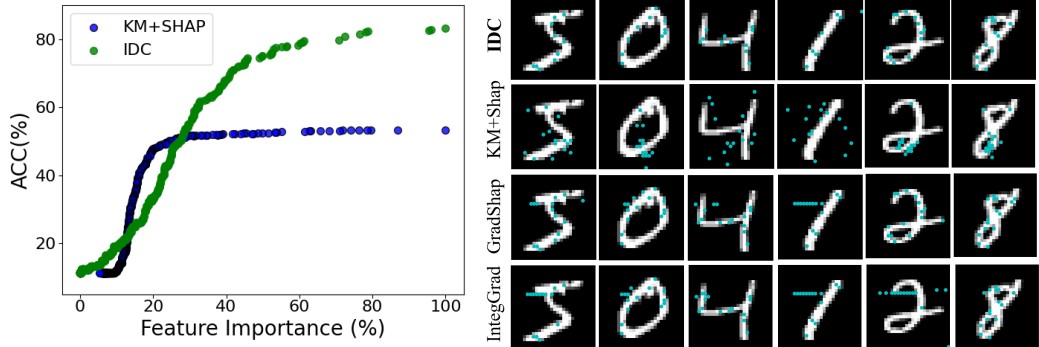

Figure 3: **Left**: Faithfulness plot of the proposed method (green) and $K$-Means+SHAP (blue) on MNIST$_{10K}$ subset. Accuracy drop and feature importance are well correlated for our approach $0.96$ (see green dots) while less correlated for SHAP features with $K$-means clustering $0.79$ (see blue dots). Furthermore, notice that $K$-means accuracy only reaches $53\%$ while our method - $83\%$. **Right**: Features selected by different explainers on MNIST$_{10K}$. The features are learned during clustering training as proposed by our approach (top), features selected by SHAP explainer with $K$-means predictor (KM-SHAP), features obtained from the Gradient-SHAP explainer with trained clustering model (GradSHAP) and features predicted by Integrated Gradients explainer (bottom).

### 6.2.3 CLUSTERING IMAGE BENCHMARKS

**MLP architecture**   In this experiment, we train two versions of our model for each dataset. The first version learns clustering assignments without local gates, so the AE is fed raw input samples with $D = 784$. The second version is our full model with a gating network that reduces the number of non-zero gates to the minimum. Both models are trained on MNIST$_{60K}$ dataset samples with the same hyperparameters. The Encoder consists of 4 linear layers with batch normalization and ReLu non-linearity with hidden layers of sizes $[512, 512, 2048]$. The Decoder has the same but reversed structure. In addition, the clustering head has a single hidden layer of size 2048. Table 3 demonstrates how the gating network contributes to accurate clustering and selects a small subset of features.

**CNN architecture**   We conducted an additional experiment on the CIFAR10 dataset. The data has $3 \times 32 \times 32$ features, 60K samples, and 10 clusters. We modified the AE module to be a mirrored ResNet18 model with convolutions. We do not apply any additional augmentations during the training. The results are presented in Table 4. Despite not being designed for vision (avoiding domain-specific augmentations), our model leads to competitive results while using much fewer features. We run the experiment 10 times for fair comparison and pick the results with the lowest training loss as suggested in the compared works (Peng et al., 2022; Jiang et al., 2016) and results of other models borrowed from (Peng et al., 2022).

### 6.2.4 REAL WORLD DATA

This section evaluates our method using six real tabular datasets commonly used for evaluating feature selection schemes. The datasets are collected from different biological domains and often contain more features than samples. Such a regime of high dimensions and a low sample size is highly challenging for supervised (Singh et al., 2023; Liu et al., 2017) or unsupervised models (Abid

Table 3: Clustering evaluation and the number of features selected (last column) by using our method on MNIST$_{60K}$ and FashionMNIST datasets. The proposed model improves clustering accuracy by using only about 16 features out of 784 on MNIST$_{60K}$ and about 69 features on FashionMNIST datasets. At the first row we present clustering evaluation of IDC on the non-seen test set. The model is trained on the MNIST$_{60K}$ and tested on unseen MNIST test (10K samples)

| Dataset | Method | ACC↑ | ARI↑ | NMI↑ | $|\mathcal{S}|$↓ |
|---|---|---|---|---|---|
| MNIST (test) | IDC | 83.6 | 77.0 | 80.3 | 14.9 |
| MNIST$_{60K}$ | IDC (w/o gates) | 81.1 | 75.9 | 80.3 | 784 |
| | IDC | **87.9** | **82.8** | **85.1** | 15.81 |
| FashionMNIST | IDC (w/o gates) | 61.0 | **49.3** | 62.7 | 784 |
| | IDC | **61.9** | 49.1 | **63.3** | 68.6 |

Table 4: Clustering performance on CIFAR10 dataset. IDC model selects 586 features (on average) out of 3,072.

| Model | ACC↑ | ARI↑ | NMI↑ |
|---|---|---|---|
| IDC | 25.01 | **6.16** | **11.96** |
| TELL(Peng et al., 2022) | **25.65** | 5.96 | 10.41 |
| VaDE(Jiang et al., 2016) | 20.87 | 3.95 | 7.20 |
| NMF(Cai et al., 2010) | 19.68 | 3.21 | 6.20 |

Table 5: Ablation study on MNIST$_{60K}$ dataset

| Model | ACC↑ | ARI↑ | NMI↑ |
|---|---|---|---|
| IDC | **87.9** | **82.8** | **85.1** |
| IDC w/o $\mathcal{L}_{reg}$ | 85.9 (-2.0) | 81.2(-1.6) | 84.7 (-0.4) |
| IDC w/o latent denoising | 86.5 (-1.4) | 80.9 (-1.9) | 83.2 (-1.9) |
| IDC w/o input denoising | 84.3 (-3.6) | 80.0 (-2.8) | 83.9 (-1.2) |
| IDC features + $K$-Means | 65.5 (-22.4) | 49.3 (-33.5) | 57.6 (-27.5) |
| IDC w/o $\mathcal{L}_{recon}$ | 18.0 (-69.9) | 2.6 (-80.2) | 4.3 (-80.8) |

Table 6: Clustering Accuracy on Real Datasets. $K$-means (KM) accuracy is borrowed from (Lindenbaum et al., 2021). $|\mathcal{S}|$ is calculated by taking the median on a batch of gates produced by LSTG and the mean value across ten experiments.

| Method / Dataset | TOX-171 | ALLAML | PROSTATE | SRBCT | BIASE | INTESTINE | PBMC-2 |
|---|---|---|---|---|---|---|---|
| KM | 41.5 ± 2 | 67.3 ± 3 | 58.1 ± 0 | 39.6 ± 3 | 41.8 ± 8 | 54.8 ± 3 | 52.37 ± 0 |
| LS+KM | 47.5 ± 1 | 73.2 ± 0 | 58.6 ± 0 | 41.1 ± 3 | 83.8 ± 0 | 43.2 ± 3 | - |
| MCFS+KM | 42.5 ± 3 | 72.9 ± 2 | 57.3 ± 0 | 43.7 ± 3 | 95.5 ± 3 | 48.2 ± 4 | - |
| SRCFS+KM | 45.8 ± 6 | 67.7 ± 6 | 60.6 ± 2 | 33.5 ± 5 | 50.8 ± 5 | 58.1 ± 10 | - |
| CAE+KM | 47.7 ± 1 | 73.5 ± 0 | 56.9 ± 0 | **62.6** ± 7 | 85.1 ± 2 | 51.9 ± 3 | 59.11 ± 6 |
| DUFS+KM | 49.1 ± 3 | **74.5** ± 1 | 64.7 ± 0 | 51.7 ± 1 | **100** ± 0 | 71.9 ± 7 | - |
| **IDC** | **50.6** ± 3 | 72.2 ± 3 | **65.3** ± 3 | 55.4 ± 5 | 95.7 ± 1 | **74.2** ± 2 | 61.56 ± 9 |
| $|\mathcal{S}|$ | 49.6 ± 8 | 307.6 ± 2 | 170.9 ± 3 | 46.7 ± 2 | 210 ± 0 | 65 ± 0 | 137.8 ± 2 |
| D / N / K | 5748 / 171 / 4 | 7192 / 72 / 2 | 5966 / 102 / 2 | 2308 / 83 / 4 | 25683 / 56 / 4 | 3775 / 238 / 13 | 17126 / 20742 / 2 |

et al., 2019). Specifically, standard clustering models fail to model the cluster assignments accurately. Table 6 presents clustering accuracy of different methods: $K$-means[4] on the full set of features, (KM), SRCFS feature selector (Huang et al., 2019) with $K$-means (SRCFS+KM), Concrete Autoencoders feature selector with $K$-means clustering on the selected features (CAE+KM) (Abid et al., 2019), DUFS feature selector (Lindenbaum et al., 2021) with $K$-means clustering (DUFS+KM), our model where the best checkpoint chosen by Silhouette score and our best model in terms of accuracy. In addition, we present the number of selected features for each dataset by our model ($|\mathcal{S}| \leq D$). Our model produces comparable clustering accuracy to the feature selection methods integrated with $K$-means and even outperforms them on three datasets. To highlight the potential of our model in bio-informatics we consider BIASE data as a use case. The data comprise of single-cell RNA sequencing (scRNA-seq). Most analysis in this domain are initiated by cell clustering using $K$-means. In this example, we dramatically improve the accuracy compared to $K$-means while identifying 210 informative genes our of more than 25K.

### 6.2.5 ABLATION STUDY

We run an ablation study to test if all reconstruction loss components are essential for model convergence. The experiment was conducted on the MNIST$_{60K}$ dataset. We run each experiment 10 times and present the results in the Table 4. As indicated by our results all of the proposed components contribute to the performance of our model.

## 7 CONCLUSIONS

We propose a deep clustering model that produces accurate cluster assignments on tabular data and predicts the informative feature set for interpretability. We evaluate the model on a synthetic dataset, commonly used deep clustering image benchmarks, and high-dimensional low sample size bio-med tabular datasets. The gating network enables built-in interpretability of our model such that the clustering is done on the sparse input but is still informative for the task. A main limitation of our scheme is dealing with correlated variables. This is a known caveat of the reconstruction loss (Abid et al., 2019) and could be alleviated by incorporating a group sparsity loss as presented by (Imrie et al., 2022). We hope our work will benefit scientists in the bio-med domain.

---

[4]Implemented here: https://scikit-learn.org/stable/modules/generated/sklearn.cluster.KMeans.html

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

## A  IMPLEMENTATION DETAILS

We implement our model in Pytorch and run experiments on a Linux server with NVIDIA GeForce GTX 1080 Ti and Intel(R) Core(TM) i7-8700 CPU @ 3.20GHz. We share our code on Github[5]. Our model has two main versions: the first includes only the autoencoder and clustering head and a learnable global gating matrix, denoted in the experiments by **IDC** (w/o gates). The second version adds a gating network to the first one, which is the complete proposed model. Every part presented in Fig. 1 is trained with a separate optimizer and learning rate.

## B  MODEL ARCHITECTURE

The model is trained with a single hidden layer in the Clustering Head and in the Gating NN. For Encoder we use hidden layers of sizes [512,512,2048] and the output dimension equals to the number of clusters. The Decoder is a mirrored version of encoder.

## C  TRAINING SETUP

We train all models with a two-stage approach - we train Encoder, Decoder, and Gating NN in the first stage and then train Clustering Head in the second stage. We use Adam optimizer for modules (Encoder, Decoder, Gating NN) with learning rate $1e-3$ and learning rate $1e-2$ for Clustering Head. To train global gates matrix with use SGD optimizer with learning rate $1e-1$. For small sample size datasets we increase the number of epochs.

For interpretabiltiy experiments, we train $K$-means [6] and TELL (Peng et al., 2022) [7]. Both methods are trained without additional augmentations for fair comparison to our method with the provided default experimental parameters.

Table 7: The number of epochs and batch size for different datasets.

| Dataset | Epochs Stage 1 | Epochs Stage 2 | Batch size |
|---|---|---|---|
| Synthetic | 50 | 2000 | 800 |
| MNIST$_{60K}$ | 300 | 600 | 256 |
| MNIST$_{10K}$ | 300 | 700 | 100 |
| FashionMNIST | 100 | 500 | 256 |
| TOX-171 | 10000 | 2000 | 171 (full) |
| ALLAML | 10000 | 1000 | 72 (full) |
| PROSTATE | 10000 | 1000 | 102 (full) |
| SRBCT | 2000 | 1000 | 83 (full) |
| BIASE | 2000 | 4000 | 301 (full) |
| INTESTINE | 4000 | 2000 | 238 (full) |
| PBMC-2 | 100 | 200 | 256 |

## D  MAXIMUM CODING RATE MODIFICATION

For simplicity in the presentation of our work, we slightly modify the original formula of the Coding Reduction Rate. As presented in the paper we use the next formula for the gating network optimization:

$$\mathcal{L}_{gtcr} = -\mathbb{E}_{\mathcal{Z}}\Big[\frac{1}{2} \cdot \text{logdet}(\mathbf{I} + \lambda_1 \cdot \mathbf{z}_i^T \mathbf{z}_i)\Big] \tag{7}$$

where is defined by $\lambda_1 = d_{input} \cdot (B * \epsilon)$, where $\epsilon$ is the the coding error hyper parameter, $B$ is a batch size and $d_{input}$ is the dimension of the input sample. Similarly, for the clustering loss term represented by:

$$\mathcal{L}_{head} = \sum_{k=1}^{K} \frac{1}{2} \cdot \text{logdet}\big[\mathbf{I} + \lambda_2 \cdot \mathbb{E}_{\boldsymbol{h}_i \in \mathbf{H}_k}(\boldsymbol{h}_i^T \boldsymbol{h}_i)\big], \tag{8}$$

---

[5]the files are in the supplementary material for this submission and will be shared on Github later

[6]https://scikit-learn.org/stable/modules/generated/sklearn.cluster.KMeans.html

[7]The implementation was found here: https://github.com/XLearning-SCU/2022-JMLR-TELL/tree/main, accessed on 2023-09-28.

we use $\lambda_2 = d_{emb} \cdot (B * \epsilon)$ where the $d_{emb}$ is the dimension of the embedding vector for each sample. We use the same $\epsilon$ for both loss terms in our experiments.

# E  REGULARIZATION TERM

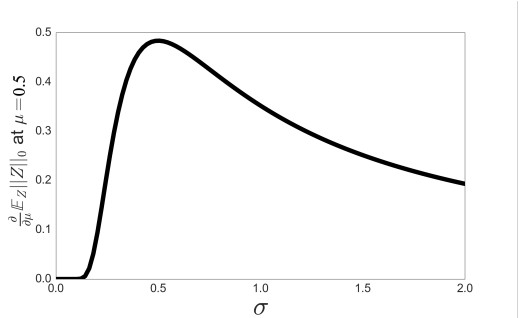

Figure 4: The value of $\frac{\partial}{\partial \mu} \mathbb{E}_Z ||\boldsymbol{Z}||_0|_{\mu=0.5} = \frac{1}{\sqrt{2\pi\sigma^2}} e^{-\frac{1}{8\sigma^2}}$ for $\sigma = [0.001, 2]$.

The leading term in our regularizer is expressed by :

$$
\begin{aligned}
\mathbb{E}_Z ||\boldsymbol{Z}||_0 &= \sum_{d=1}^{D} \mathbb{P}[z_d > 0] = \sum_{d=1}^{D} \mathbb{P}[\mu_d + \sigma\epsilon_d + 0.5 > 0] \\
&= \sum_{d=1}^{D} \{1 - \mathbb{P}[\mu_d + \sigma\epsilon_d + 0.5 \leq 0]\} \\
&= \sum_{d=1}^{D} \{1 - \Phi(\frac{-\mu_d - 0.5}{\sigma})\} \\
&= \sum_{d=1}^{D} \Phi\left(\frac{\mu_d + 0.5}{\sigma}\right) \\
&= \sum_{d=1}^{D} \left(\frac{1}{2} - \frac{1}{2}\,\mathrm{erf}\left(-\frac{\mu_d + 0.5}{\sqrt{2}\sigma}\right)\right)
\end{aligned}
$$

To tune $\sigma$, we follow the suggestion in Yamada et al. (2020). Specifically, the effect of $\sigma$ can be understood by looking at the value of $\frac{\partial}{\partial \mu_d} \mathbb{E}_Z ||\boldsymbol{Z}||_0$. In the first training step, $\mu_d$ is 0. Therefore, at initial training phase, $\frac{\partial}{\partial \mu_d} \mathbb{E}_Z ||\boldsymbol{Z}||_0$ is close to $\frac{1}{\sqrt{2\pi\sigma_d^2}} e^{-\frac{1}{8\sigma_d^2}}$. To enable sparsification, this term (multiplied by the regularization parameter $\lambda$) has to be greater than the derivative of the loss with respect to $\mu_d$ because otherwise $\mu_d$ is updated in the incorrect direction. To encourage such behavior, we tune $\sigma$ to the value that maximizes the gradient of the regularization term. As demonstrated in Fig. 4 this is obtained when $\sigma = 0.5$. Therefore, we keep $\sigma = 0.5$ throughout our experiments unless specifically noted.

# F  DATASETS PROPERTIES AND REFERENCES

In Table F we add the references of the datasets used in the experiments.

# G  TRAIN LOSS AUGMENTATIONS

In addition to the loss presented in Section 4.2 we exploit the next dataset-agnostic augmentations during model training. The first one is the standard reconstruction loss that is calculated between

Table 8: Properties and references for the dataset used in the experiments.

| Dataset | Features | Samples | Clusters | Reference |
|---------|----------|---------|----------|-----------|
| MNIST$_{10K}$ | 784 | 10,000 | 10 | lec |
| MNIST$_{60K}$ | 784 | 60,000 | 10 | lec |
| FashionMNIST$_{60K}$ | 784 | 60,000 | 10 | git |
| TOX-171 | 5,748 | 171 | 4 | jun |
| ALLAML | 7,192 | 72 | 2 | jun |
| PROSTATE | 5,966 | 102 | 2 | jun |
| SRBCT | 2,308 | 83 | 4 | Khan et al. (2001) |
| BIASE | 25,683 | 56 | 4 | Biase et al. (2014); Fan et al. (2015); Sato et al. (2009); Pollen et al. (2014) |
| INTESTINE | 3,775 | 238 | 13 | Biase et al. (2014); Fan et al. (2015); Sato et al. (2009); Pollen et al. (2014) |
| PBMC-2 | 17,126 | 20,742 | 2 | Zheng et al. (2017) |
| CIFAR10 | 3,072 | 60,000 | 10 | Krizhevsky et al. (2009) |

input samples and reconstructed samples. Input denoising is based on Vincent et al. (2008) and latent denoising on Doi et al. (2007):

- *Clean reconstruction loss*, $||f_{\theta_3} \circ f_{\theta_2}(\boldsymbol{x}_i) - \boldsymbol{x}_i||_1$, which measures the deviation of estimated $\hat{\boldsymbol{x}}_i$ from the input sample $\boldsymbol{x}_i$.
- *Denoising reconstruction loss* Vincent et al. (2008), $||f_{\theta_3} \circ f_{\theta_2}(\boldsymbol{x}_i \odot m_{rand}) - \boldsymbol{x}_i||_1$, where $m_{rand} \in \{0, 1\}^D$ is a random binary mask generated for each sample $\boldsymbol{x}_i$. We generate a mask such that about $90\% - 99\%$ of the input features are multiplied by zero value, which indicates that the gate is closed. The loss pushes the method to pay less attention to unnecessary features for the reconstruction.
- *Latent denoising reconstruction loss*, $||f_{\theta_3}(\boldsymbol{h}_i \cdot h_{noise}) - \boldsymbol{x}_i||_1$, where $h_{noise} \sim \mathcal{N}(1, \sigma_h)$ is a noise generated from a normal distribution with mean one and scale $\sigma_h$ which is a dataset-specific hyperparameter Doi et al. (2007). This term aims to improve latent embedding representation by small perturbation augmentation to treat small sample-size datasets.

## H  MODEL TRAINING SCALABILITY

In Figure H we show the training time as a function of number of data samples. It could be seen that training time scales linearly with an increase in dataset length.

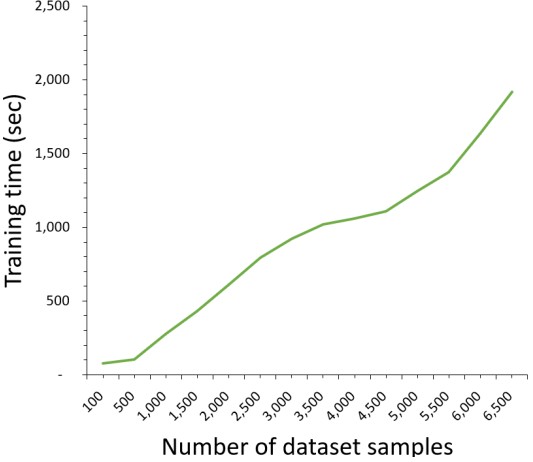

Figure 5: Training time in seconds measured for different numbers of samples.

## I  SYNTHETIC DATASET GENERATION

The dataset consists of three informative features $x_i[j] \in [-1, 1], j = 1, .., 3$ for each sample $\boldsymbol{x}_i$ and is generated as isotropic Gaussian blobs [8] with standard deviation of each cluster of 0.5. The detailed

---

[8]https://scikit-learn.org/stable/modules/generated/sklearn.datasets.make_blobs.html

description of the dataset generation could be found in I. Then we add ten nuisance background features with values drawn from $\mathcal{N}(0, \sigma_n^2)$ (with $\sigma_n = 0.1$) resulting in 13 total features. The samples are equally distributed between 4 clusters, with $\sim 800$ samples in each cluster. Given the first two dimensions $\{x[1], x[2]\}$, only 3 clusters are separable, and the same property holds for dimensions pair $\{x[1], x[3]\}$.

