# OpenReview forum: "Interpretable Deep Clustering"
_ICLR.cc/2024/Conference — Submitted to ICLR 2024_

### Official Review · Reviewer_viN8 · 2023-10-16

**Soundness:** 3 good
**Presentation:** 2 fair
**Contribution:** 2 fair
**Rating:** 5
**Confidence:** 4

**Summary:**

The author proposed a new deep-learning framework for tabular data that predicts interpretable cluster assignments at the instance and cluster levels. They also  validated the performance in both synthetic and tabular biological datasets. However, this article overall did not meet the requirements of ICLR.

**Strengths:**

1. The paper is easy to follow
2. The experiments are comprehensive

**Weaknesses:**

1. The motivation is not clear, I don not understand what is interpretable clustering model and why we need  interpretable clustering model.
2. The overall method is just the combination of existing approaches, the novelty is limited
3. I don not agree with the authors the empirical diversity, faithfulness and uniqueness can represent  interpretability
4. The manuscript was not well prepared. It contains obvious typos, such as the citation "?" in third line of page six
5. The improvement in real-world dataset is not significant

**Questions:**

1. The motivation is not clear, I don not understand what is interpretable clustering model and why we need  interpretable clustering model.
2. The overall method is just the combination of existing approaches, the novelty is limited
3. I don not agree with the authors the empirical diversity, faithfulness and uniqueness can represent  interpretability
4. The manuscript was not well prepared. It contains obvious typos, such as the citation "?" in third line of page six
5. The improvement in real-world dataset is not significant

---

> ### Author Response · Authors · 2023-11-17
> **Response to Reviewer viN8**
>
> We thank the reviewer for the time and effort spent in this review and for appreciating that the paper is easy to follow and that the experiments are comprehensive. Below, we address all comments raised by the reviewer.
>
> P1 Motivation of interpretable clustering-
>
> The second paragraph in the introduction motivates our interpretable clustering method. To elaborate on the problem, we offer the following paragraph:
>
> The main goal of clustering methods is to partition the data into semantically related groups. In some use cases, it is sufficient to assess the separability of the clusters to evaluate the clustering quality. However, clusters are used in many applications for downstream tasks, such as scientific discovery, decision-making, medical analysis, and more. In such cases, cluster assignment interpretations are crucial for annotating the clusters manually. Cluster interpretability is also vital for quality assessment, model validation, and gaining trust in practitioners.
>
> For example, let’s consider the task of cell clustering (also mentioned in the introduction). Common practice is to use clustering and then manually analyze each cluster to identify which cell types it represents. Cell annotation is vital for downstream tasks, such as drug discovery, personalized treatment, and automated diagnosis and prognosis. In these cases, identifying cluster-level and sample-level informative features is crucial for labeling the clusters and validating the separation. Another example is a medical setting where we seek groups of patients with similar characteristics. In such applications, doctors seek to understand what makes each cluster unique in terms of demographics, genetic information, and diagnostics.
>
>
>
>
> P2- Novelty of technical elements-
>
> We appreciate the feedback regarding the perceived limited novel elements in our framework. We want to address this comment by highlighting the contributions of our work, which goes beyond the technical algorithmic components.
>
> The novelty of our work could be briefly summarized as follows:
> Formulate an interpretable clustering task that highlights sample and cluster-level informative features.
> Generalize the unsupervised feature selection with clustering problem to the dynamic sample-wise setting.
> Present an end-to-end neural network-based solution to the proposed interpretable clustering problem. Besides integrating existing methodologies, we have also introduced new (or modified) components. These include (1) a pre-training augmentation scheme that does not require domain-specific knowledge (as used in vision, audio, NLP), see full details Appendix G. (2) The gates coding loss (Eq. 4), which encourages uniqueness of the sample-specific gates. (3) Combining a Gumbel-softmax clustering head with the maxima coding rate loss (eq. 6).
> Extensive empirical evaluation of the model's ability to explain its predictions and perform accurate clustering using synthetic and real-world image and tabular data.
>
> As a final remark about novelty, we want to emphasize that many well-celebrated papers in the machine-learning community rely on the combination and adaptation of existing schemes; examples of such works include [1], [2], [3]. We believe that this should be encouraged by the community. The main novelty here is that the presented problem and solution were never addressed.
>
> [1] He et al.” Deep Residual Learning for Image Recognition, 2016” used skip connections (already an existing technique) to enable gradient propagation in deep layers.
>
> [2] Vaswani et al. “Attention Is All You Need”- They used the attention mechanism (commonly applied for text) to image data.
>
> [3] Devlin et al.” BERT: Pre-training of Deep Bidirectional Transformers for Language Understanding”- used existing transformer techniques but trained to predict text conditioned on both left and right context in all layers.

---

> ### Author Response · Authors · 2023-11-17
> **Response to Reviewer viN8 part 2**
>
> P3 Interpretability metrics-
>
> The main goal of interpretability in ML models is to make the predictions more understandable and transparent for humans; diversity, faithfulness, and uniqueness do not “represent” interpretability; instead, they were proposed in [1,2] as objective metrics to compare different qualities of interpretability models.
> Diversity measures how well the interpretability model captures the differences between groups (clusters) and is not overly focused on specific features globally.
> Faithfulness evaluates how well the explanations coincide with the mechanism that drives the decision of the ML model.
> Uniqueness generalizes diversity by measuring how the interpretability model can capture different nuances within a cluster. For instance, if a specific cluster has high uniqueness values, the user can interpret that the cluster might contain a union of two (or more) subgroups with distinct characteristics.
>
> We adopt these metrics to provide quantitative comparisons between different schemes. In practice, users can use these metrics to select which method to use and analyze the quality of the clusters and clustering model. We would be happy to include other metrics that the reviewer thinks are more useful for measuring the qualities of our model.
>
> These metrics were proposed in prior work to evaluate model interpretability quality [1,2].
> We agree with the reviewer that interpretability is a broad concept with different implementations. In this work, we do not focus on the model interpretability in terms of its architecture. This work focuses on predictions interpretability, and we follow previous works [1,2] that proposed diversity and faithfulness. In addition, since we claim that the model produces unique explanations for each sample, we propose the uniqueness metric. The intuition is that the dataset staples are generally not exact duplicates, and each sample has its uniqueness.
>
> [1] David Alvarez Melis and Tommi Jaakkola. Towards robust interpretability with self-explaining neural networks. Advances in neural information processing systems, 31, 2018.
> [2] Junchen Yang, Ofir Lindenbaum, and Yuval Kluger. Locally sparse neural networks for tabular biomedical data. In the International Conference on Machine Learning, pp. 25123–25153. PMLR, 2022.
>
> P4 Typos-
>
> Thanks for this comment; we have fixed this issue caused by a technical error. Furthermore, following this comment, we have looked for additional typos. We would be happy to correct any typos the reviewer is aware of.
> P5 Performance improvement-
> While in some datasets, clustering accuracy improvement is not significant compared to SOTA baselines; we argue that those results require careful tuning of the unsupervised feature selection method to obtain these competitive results. In contrast, our method does not require any parameter tuning. Since clustering with tabular data is challenging, several works have shown that NN-based solutions tend to be inferior to unsupervised feature selection models followed by K-means [1] [2]. When we compare our result to K-means,  our method leads to an average improvement of more than 37% in clustering accuracy. Furthermore, the main advantage of our method is interpretability, serving as the only method that provides sample-level and cluster-level informative features while improving the SOTA clustering capabilities on most of the datasets.
>
> [1] Abrar et al. Effectiveness of Deep Image Embedding Clustering Methods on Tabular Data, 2023.
>
> [2] Solorio-Fernández et al. A review of unsupervised feature selection method, 2019
>
> We thank the reviewer again for these constructive comments that helped improve our paper.
> We would happily provide additional information if the reviewer still has any open issues or questions.

---

> > ### Comment · Reviewer_viN8 · 2023-11-20
> > **Raising rating**
> >
> > Thanks for you careful responses.  I agree with the authors the interpretable clustering method is important for many bioinformatics tasks. We did several works about image clustering in the past years. Actually, the recent image clustering methods, such as SCAN[1],  GCC[3] and TCC[4], perform extremely well on many image datasets that are more challenging than MNIST.  In my opinion, the proposed method is relatively outdated. I understand that applying these state-of-the-art approaches to biological information takes time, so I will raise my rating to 5: marginally bellow the acceptance threshold.
> >
> > [1]Van Gansbeke W, Vandenhende S, Georgoulis S, et al. Scan: Learning to classify images without labels[C]//European conference on computer vision. Cham: Springer International Publishing, 2020: 268-285.
> >
> > [2]Zhong H, Wu J, Chen C, et al. Graph contrastive clustering[C]//Proceedings of the IEEE/CVF international conference on computer vision. 2021: 9224-9233.
> >
> > [3]Shen Y, Shen Z, Wang M, et al. You never cluster alone[J]. Advances in Neural Information Processing Systems, 2021, 34: 27734-27746.

---

> ### Author Response · Authors · 2023-11-21
> **Response to Reviewer viN8**
>
> We want to thank the reviewer for acknowledging the importance of the addressed problem and for raising the score. Thanks for pointing out these image clustering models. Indeed, the methods presented in [1,2,3] can provide accurate clustering assignments on image data. However, it is important to note that (1) they do not include any interpretability component and, therefore, do not address our primary objective. (2) can not be easily adapted to tabular data while maintaining high clustering capabilities. Specifically, all these methods use self-supervision and, more specifically, rely on data augmentation to create positive and negative pairs. Creating meaningful augmentations is feasible in vision because we understand the different properties of the domain. For example, we know that small color changes, rotations, translations, and rescalings do not change the “label” of the image and can be used to create positive pairs. In tabular data, we do not have this privilege, and augmentations are restricted to additive noise and masking, which, in practice, do not lead to significant performance gains [4,5,6].
> Nonetheless, following the reviewer's suggestion, we have evaluated the GCC [2,6] method on two datasets (1) MNIST train split (60K samples, 784 features, 10 clusters), (2) a biomed PBMC dataset (20,742 samples, 17,126 features, 2 clusters). We used random zero masks for augmentation, similar to what we used in our paper. The architecture is an MLP with layers: [input, 512, 512, 2048, 512] as a backbone. We train GCC on these datasets during 300 epochs for the MNIST dataset and 100 epochs for the PBMC dataset; we present here the clustering accuracy results:
> | Dataset | ACC (*GCC*)  | ACC (*Our*) |
> |----------|----------|----------|
> | MNIST$_{60K}$ |  63.52 | **87.90** |
> | PBMC-2 | 52.31 | **61.56** |
>
> We thank the reviewer again for these constructive comments that helped improve our paper.
> We would be happy to provide additional information if the reviewer still has any open issues or questions.
>
> [1] Van Gansbeke W, Vandenhende S, Georgoulis S, et al. Scan: Learning to classify images without labels[C]//European conference on computer vision. Cham: Springer International Publishing, 2020: 268-285.
>
> [2] Zhong H, Wu J, Chen C, et al. Graph contrastive clustering[C]//Proceedings of the IEEE/CVF international conference on computer vision. 2021: 9224-9233.
>
> [3] Shen Y, Shen Z, Wang M, et al. You never cluster alone[J]. Advances in Neural Information Processing Systems, 2021, 34: 27734-27746.
>
> [4] Mai et al. Understanding the limitations of self-supervised learning for tabular anomaly detection. Arxiv, 2023
>
> [5] Hajiramezanali et al. STab: Self-supervised Learning for Tabular Data. NeurIPS 2022 Workshop on Table Representation Learning.
>
>
> [6] https://github.com/mynameischaos/GCC

---

### Official Review · Reviewer_AryA · 2023-10-31

**Soundness:** 2 fair
**Presentation:** 3 good
**Contribution:** 2 fair
**Rating:** 6
**Confidence:** 4

**Summary:**

The paper introduces an Interpretable Deep Clustering model that predicts an informative feature set for improved interoperability. Leveraging a self-supervised reconstruction task, the method employs stochastic gates to learn instance-level feature selection, which can be extended to the cluster-level form. The two-stage training process involves losses encompassing reconstruction errors and various constraint terms. Overall, the paper offers valuable insights and demonstrates its superiority in terms of performance and interoperability.

**Strengths:**

1. The paper is well-structured, featuring clear logic and technical explanations that allow readers to easily follow the authors' design. Additionally, the manuscript is well-written overall, demonstrating proficient English grammar and adhering to a formal writing style that aligns with academic standards for technical manuscripts.
2. The proposed method is technically sound and demonstrates impressive performance on both synthetic and real datasets.
3. The paper's approach to designing a clustering model with a focus on interoperability offers an intriguing perspective.

**Weaknesses:**

1. The paper's novelty appears to be somewhat incremental, as it combines existing unsupervised feature selection (stochastic gates) with deep clustering, lacking significant novel elements.
2. The main design of the model lacks a theoretical guarantee. For instance, the reasoning behind choosing an autoencoder (AE) over other self-supervised tasks, such as contrastive learning, requires clarification.
3. The method's generalizability to unseen data is not adequately explained. Eq. (6) suggests high computational complexity, necessitating a discussion on the complexity for better understanding.
4. The experiment comparison seems biased. While the proposed method employs strong feature transformation by DNN, competitors like k-means do not. Hence, a fair comparison with state-of-the-art deep clustering models is essential.
5. It would be beneficial to discuss the model's performance on a large-scale dataset to provide a comprehensive evaluation.
6. The subscripts in Eq. (6) should be carefully reviewed for accuracy.

**Questions:**

Please see the cons for details.

---

> ### Author Response · Authors · 2023-11-17
> **Response to Reviewer AryA**
>
> We thank the reviewer for the time and effort spent in this review and for appreciating our writing and the quality of our results. Below, we address all comments raised by the reviewer.
>
> P1- Technical novelty-
> We appreciate the feedback regarding our framework's perceived lack of significant novel elements. We want to address this comment by highlighting the contributions of our work, which goes beyond the technical algorithmic components.
>
> The novelty of our work could be briefly summarized as follows:
> Formulate an interpretable clustering task that highlights sample and cluster-level informative features.
> Generalize the unsupervised feature selection with clustering problem to the dynamic sample-wise setting.
> Present an end-to-end neural network-based solution to the proposed interpretable clustering problem. Besides integrating existing methodologies, we have also introduced new (or modified) components. These include (1) a pre-training augmentation scheme that does not require domain-specific knowledge (as used in vision, audio, NLP), see full details Appendix G. (2) The gates coding loss (Eq. 4), which encourages uniqueness of the sample-specific gates. (3) combining a Gumbel-softmax clustering head with the maximal coding rate reduction loss (eq. 6).
> Extensive empirical evaluation of the model's ability to explain its predictions and perform accurate clustering using synthetic and real-world image and tabular data.
>
> As a final remark, we want to emphasize that many well-celebrated papers in the community rely on the combination and adaptation of existing schemes; examples include [1], [2], [3]. We believe that this should be encouraged by the community. The main novelty is that the presented problem and solution were never addressed.
>
> [1] He et al.,” Deep Residual Learning for Image Recognition, 2016” used skip connections (already an existing technique) to enable gradient propagation in deep layers.
>
> [2] Vaswani et al. “Attention Is All You Need”- used the attention mechanism (commonly applied to text) for image data.
>
> [3] Devlin et al.” BERT: Pre-training of Deep Bidirectional Transformers for Language Understanding”- used existing transformer techniques but trained to predict text conditioned on both left and right context in all layers.
>
> P2- AE:
>
> Indeed, contrastive learning is a powerful tool for self-supervision but requires data augmentations to create “positive pairs.” In vision, audio, and NLP, we can use domain knowledge to design augmentations that preserve each sample's semantic information. Developing such augmentations for tabular data is much more challenging. The main goal of our work is to present a self-supervised scheme for tabular data (for instance, genomic data). Therefore, we decided to use an autoencoder (AE), which was demonstrated effective for clustering and unsupervised feature selection in prior work [4,5,6]. It is important to note that we did not use a standard AE; instead, we have introduced perturbation to the input and latent pairs (see full details Appendix G), which strengthens the ability of our model to identify informative features. We will clarify this in the main text.
> Regarding the theoretical analysis, we are currently working on analyzing the feature selection capabilities of our model, but this requires a particular data model and, therefore, does not fit the message of the current paper.
> [4] Xie, et al. "Unsupervised deep embedding for clustering analysis" 2016.
>
> [5] Sokar et al. "Where to pay attention in sparse training for feature selection?" 2022.
>
> [6] Li et al. "Reconstruction-based unsupervised feature selection" 2017.
>
>
>
> P3 - Generalizability-
>
> Since our method is fully parametric, it offers generalization capabilities. Our model can predict cluster assignments and informative features for samples not seen during training. We have verified the generalization capabilities of our model in the MNIST experiment. We have predicted the assignments for 10,000 unseen samples. As indicated by the results presented in Table 3, on unseen samples, our model leads to similar clustering accuracies and uses the same amount of selected features. We will clarify this in the text.

---

> > ### Author Response · Authors · 2023-11-17
> > **Response to reviewer AryA part 2**
> >
> > P4 Computation complexity of Eq.6-
> > Assuming a batch of $n$ samples, each with latent dimension $m$, we calculate the correlation matrix by normalizing the embeddings as $O(m^2 \cdot n)$. Then, to calculate the logarithm of the determinant, we need $O(min(n,m)^3)$ operations. Assuming we have $K$ clusters, we perform this calculation for every cluster, and overall, we need $K\cdot(O(m^2\cdot n)+O(min(n,m)^3))$ operations.
> > In practice, our model can scale to large datasets due to the parallelization of GPU and since we can use small batches. Appendix H provides an empirical evaluation of the training time required by our model vs. the sample size.
> > P5 Empirical comparison to NN-based models.
> > Table 4 compares the SOTA NN-based clustering methods on the CIFAR10 dataset. For the real-world tabular data, most deep learning-based clustering schemes typically underperform (see [7]), and unsupervised feature selection schemes followed by K-means usually lead to more accurate cluster assignments (see [8]). It is important to note that CAE (in Table 6) is based on an AE and can also learn complex feature relations using a NN. If the reviewer knows any existing NN scheme for unsupervised feature selection and clustering, we would happily include it as a baseline.
> > [7] Abrar et al. Effectiveness of Deep Image Embedding Clustering Methods on Tabular Data, 2023.
> > [8] Solorio-Fernández et al. A review of unsupervised feature selection method, 2019
> >
> > P6 large datasets-
> > Most of our evaluations focus on datasets where the number of features is larger than the number of samples. In such a setting, noisy features can obscure the clusters, and finding informative features is vital. As demonstrated in Table 6, in these datasets, removing features improves clustering accuracy. We have also evaluated larger datasets such as MNIST and CIFAR-10. Our method could also scale to larger datasets since training is performed in small batches.
> >
> > P7 Eq. 6-
> > We will add the following paragraph to clarify this equation:
> > Given a batch of normalized latent embeddings, we predict for each sample its cluster assignment. Then, for each cluster, we multiply the stacked vectors by their transpose to calculate the correlation matrix, which is also a covariance matrix since the vectors are normalized. Then, we add an identity matrix and calculate the Log determinant of the result. Finally, we sum up the terms from each cluster category k to obtain the loss value. Our goal is to minimize the sum of the volume of each cluster to compress them.
> >
> >
> >
> > We thank the reviewer again for these constructive comments that helped improve our paper.
> > We would happily provide additional information if the reviewer still has any open issues or questions.

---

### Official Review · Reviewer_6xNv · 2023-10-31

**Soundness:** 4 excellent
**Presentation:** 4 excellent
**Contribution:** 3 good
**Rating:** 8
**Confidence:** 4

**Summary:**

The authors develop a novel deep clustering and feature selection method. The proposed model employs a two-stage approach. In the first stage, a Gating Network and an autoencoder are used for self-supervised learning of latent representations and sample-level informative features. In the second stage, a clustering head is trained to predict cluster assignments based on these latent embeddings. The model aims to provide both instance-level and cluster-level explanations by selecting a subset of features that are most informative for each cluster. The paper validates the model's performance through a series of experiments conducted on synthetic datasets, including well-known benchmarks like MNIST, FashionMNIST, and CIFAR10. The experiments show the model outperforms other clustering strategies while maintaining interpretability. The paper also includes ablation studies to understand the impact of various components of the model on its performance.

**Strengths:**

Comprehensive Experiments: the paper conducts a wide range of experiments across multiple datasets, including synthetic datasets, MNIST, FashionMNIST, and CIFAR10. A exploration of the time it takes to run the the method according to dataset size is also provided.

Interpretability Focus: one of the key strengths of the paper is its focus on interpretability. The model aims to provide both instance-level and cluster-level explanations, which is crucial for understanding the model's decisions and could be particularly useful in sensitive applications.

Innovative Approach: the paper proposes a novel two-stage approach that combines self-supervised learning for feature selection and a clustering head for cluster assignment. This is an innovative way to tackle the problem and could inspire future research in this area.

Ablation Studies: the paper includes ablation studies to understand the impact of various components of the model, confirming that are components of the method are, indeed, relevant to its performance.

**Weaknesses:**

The paper addresses everything I would expect in a clustering paper, especially with the interpretability focus.
Perhaps the only weakness would be the lack of a deeper discussion on interpretability and its different perspectives in machine learning, but that does not decrease the quality of the paper.

**Questions:**

-

---

> ### Author Response · Authors · 2023-11-17
> **Response to Reviewer 6xNv**
>
> We thank the reviewer for the time and effort spent in the review and for appreciating our new method and the reported results. Bellow, we address all comments raised by the reviewer
>
>
> Thanks for your comment about interpretability; we added the following paragraph to deepen the discussion on interpretability:
>
> Interpretability in machine learning refers to the ability to understand and explain the predictions and decisions made by the predictive models. It's critical for properly deploying machine learning systems, especially in applications where transparency and accountability are essential. Interpretability comes in different forms, to name a few: interpretable model structure, identifying feature importance for model predictions, visualization of data, and generation of explanations for the prediction. In this work, we aim to design a model that achieves interpretability by sample-wise feature selection and generating cluster-level interpretations of model results. This type of interpretability is crucial for biomedical applications, for example, when seeking marker genes that are “typical” for different clusters in high throughput biological measurements.

---

### Meta-Review · Area_Chair_m6Ny · 2023-12-12

**Metareview:**

This paper addresses the challenge of predicting interpretable cluster assignments (in tabular data). Toward this, the authors propose a novel deep-learning framework incorporating a self-supervised feature selection procedure and a model that predicts cluster assignments along with a gate matrix for cluster-level feature selection. The authors argue that the proposed method demonstrates reliable cluster assignment predictions in both synthetic and tabular biological datasets, offering insights into the driving features for each sample and cluster.

Three reviewers evaluated the paper, with two recommending acceptance and one suggesting rejection. Despite author rebuttals and subsequent internal discussions with AC, a consensus among the reviewers could not be reached.

While the abstract suggests a model specialized for tabular data, in reality, the proposed method, which does not rely on self-supervised learning based on image-specific augmentations, just can be applied to tabular data. Hence, there is no reason why the proposed method cannot be applied to image data, and experiments are being conducted on both tabular and image data. Even without considering technical details, I believe experimentally that this paper fails to meet the high bar set by ICLR for the following reasons.
- The paper is validated on six tabular datasets, and according to Tabular 6, the proposed method not only shows the best performance in only three out of the six datasets but also does not exhibit statistically significantly superior performance, as the performance overlaps within the confidence interval. Therefore, I do not consider the proposed method to demonstrate particularly strong performance in tabular data.
- The proposed method is being validated in the image domain, but, as pointed out by reviewers, a performance comparison with models specialized for the image domain is necessary. Although a comparison experiment was conducted on one model and two datasets through rebuttal, it still appears to be insufficient. In the mentioned SCAN [1], GCC [2], and TCC [3] papers, comparisons are made with a much larger number of baselines on a broader set of image datasets, including ImageNet. Additionally, experiments added by the authors in the rebuttal show significant discrepancies compared to the values reported in the baseline paper. It seems necessary to conduct experiments with aligned settings for a meaningful comparison.

**Justification For Why Not Higher Score:**

I do not believe the proposed method has been sufficiently experimentally validated

**Justification For Why Not Lower Score:**

n/a

---

### Decision · Program_Chairs · 2024-01-16

Reject